# cGAS and STING in Host Myeloid Cells Are Essential for Effective Cyclophosphamide Treatment of Advanced Breast Cancer

**DOI:** 10.3390/cancers17071130

**Published:** 2025-03-28

**Authors:** Yein-Gei Lai, Hao-Ting Liao, Yung-Hsiang Chen, Shih-Wen Huang, Yae-Huei Liou, Zhen-Qi Wu, Nan-Shih Liao

**Affiliations:** 1Institute of Molecular Biology, Academia Sinica, Taipei 115, Taiwan; mbyl@gate.sinica.edu.tw (Y.-G.L.); as0201263@gate.sinica.edu.tw (H.-T.L.); john_chen@hanchorbio.com (Y.-H.C.); shih-wen.huang@beigene.com (S.-W.H.); mandy19@gate.sinica.edu.tw (Y.-H.L.); alice.wu@cvhunting.com (Z.-Q.W.); 2Department of Life Sciences, National Central University, Taoyuan 320, Taiwan

**Keywords:** cyclophosphamide, cGAS, STING, type I interferon, myeloid cells, intratumoral CD8^+^ T cells

## Abstract

Mechanisms underlying the T-cell-dependent anti-cancer effect of cyclophosphamide (CTX) are not fully understood. We found that CTX chemotherapy induces long-term survival of mice bearing metastasized breast cancer with chromosomal instability. Given that CTX induces DNA damage and type I interferon (IFN-I) in vivo and that cGAS senses cytosolic double-stranded DNA, we investigated the role of the host cGAS–STING–IFN-I pathway in CTX therapy. We found that CD8^+^ T cells, the cGAS of bone marrow (BM)-derived cells, the STING of type 1 conventional dendritic cells (cDC1s), and the IFN-I response of non-cDC1 myeloid cells are essential for CTX efficacy. Moreover, the cGAS and STING of BM-derived cells positively modulate intratumoral CD8^+^ T cell populations. Our study shows that the T-cell-dependent anti-tumor effect of CTX critically involves the cGAS–STING–IFN-I axis, the IFN-I response, and the STING-independent cGAS function of host myeloid cells, which supports the deployment of CTX in treating advanced solid tumors by bypassing the often-failed IFN-I production by tumor cells due to a chronic activation of the intrinsic cGAS–STING caused by chromosomal instability.

## 1. Introduction

Cyclophosphamide (CTX) is a prodrug that has been used to treat cancer and autoimmune diseases since 1959 [1]. The bioactivation of CTX occurs in the liver via its ring oxidation by cytochrome P450 enzymes into 4-hydroxy-CTX and its tautomer, aldophosphamide. Upon uptake into the cell, aldophosphamide is converted into active acrolein and phosphoramide mustard that damages DNA. Phosphoramide mustard is a bifunctional alkylating agent that targets guanine to form DNA crosslinks, with the interstrand DNA crosslinks contributing significantly to cytotoxicity [2]. Despite killing the proliferating tumor cells, the suppression of tumor growth by CTX treatment in several immunocompetent murine models requires CD8^+^ T cells [3,4,5]. Although CTX exerts multiple immunomodulatory activities in vivo, the mechanism underlying the T-cell-dependent anti-tumor effect of CTX is not fully understood.

CTX induces the immunogenic cell death (ICD) of tumor cells [6,7], thereby stimulating the immune system via the adjuvanticity of ICD-induced damage-associated molecular patterns (DAMPs) [8]. CTX also acts directly on lymphocytes, as evidenced by the selective depletion of regulatory CD4^+^ T cells under low-dose CTX treatment (20–30 mg/kg body weight in mice) and lymphodepletion under a medium-dose regimen (100–200 mg/kg CTX in mice) [9,10]. The post-lymphodepletion recovery phase is characterized by the mobilization and expansion of dendritic cells (DCs) [7,11] and the activation and differentiation of CD4^+^ and CD8^+^ T cells [11,12,13,14,15]. Moreover, CTX induces cytokine production in vivo, such as the transient production of interleukin (IL) 6 and C-C motif chemokine ligand 2 within 4 to 48 h [13], as well as the relatively late and lasting production of type I IFN (IFN-I) from day 2 to day 10 after treatment [7,12,13,14]. Not only does the CTX-induced DC homeostasis and T cell activation promote recovery from lymphodepletion, but it also contributes to anti-tumor immunity, which requires the host response to IFN-I [7,12,13].

Double-stranded DNA (dsDNA) mislocalized in the cytosol acts as a DAMP, which is sensed by cyclic GMP-AMP synthase (cGAS), which leads to the generation of 2′3′-cyclic guanosine monophosphate–adenosine monophosphate (cGAMP) that activates the cGAS stimulator of interferon genes (STING) at the endoplasmic reticulum (ER). This activated STING undergoes tetramerization in the ER and is then translocated to the Golgi, where it serves as a signaling platform for TANK binding kinase 1 (TBK1) and the inhibitor of nuclear factor-κB (NFκB) kinase (IKK) complex that promotes the activation of interferon regulatory factor 3 (IRF3) and NFκB, as well as the subsequent induction of IFN-I and NFκB-driven inflammatory genes in the nucleus, respectively [16]. Given that CTX induces DNA damage and IFN-I production in vivo, it may therefore activate the cGAS–STING–IFN-I pathway. An early in vitro study showed that mafosfamide, a CTX analog that spontaneously degrades to 4-hydroxy-CTX, upregulates the production of IFN-I in two out of the four human breast cancer cell lines examined and that STING knockdown reduces the level of IFN-I expression [17]. However, whether and how the host cGAS–STING–IFN-I axis contributes to the anti-tumor effect of CTX in vivo remains unknown. Here, we employed a metastatic breast cancer mouse model using the EO771 cell line [18,19] that carries a high mutation burden [20] to investigate the role of host cGAS, STING, and the IFN-I response in the anti-tumor effect of CTX.

## 2. Materials and Methods

### 2.1. Mice and Cell Lines

C57BL/6JNarl (RRID:MGI:5699857) (B6) mice were purchased from the National Laboratory Animal Center (Taipei, Taiwan). *Batf*^−/−^ (Strain#013755, RRID:IMSR_JAX:013755), *IFNar1*^−/−^ (Strain#028288, RRID:IMSR_JAX:028288), *Cgas*^−/−^ (Strain#026554, RRID:IMSR_JAX:026554), *Sting^gt/gt^* (Strain#017537, RRID:IMSR_JAX:017537), *LysM-Cre* (Strain#004781, RRID:IMSR_JAX:004781), *Xcr1-Cre* (Strain#035435, RRID:IMSR_JAX:035435), *Sting^f/f^* (Strain#031670, RRID:IMSR_JAX:031670), and *IFNar1^f/f^* (Strain#028256, RRID:IMSR_JAX:028256) mouse strains were purchased from The Jackson Laboratory (JAX) (Bar Harbor, ME, USA). The *Rag2*^−/−^ and *CD11c-Cre-GFP* strains were kindly provided by Drs. Kuo-I Lin (Academia Sinica (AS), Taipei, Taiwan) and Alexander Chervonsky (University of Chicago, USA) [21], respectively. The *Il15^f/f^* strain was generated in our lab [22]. Female 7-to-9-week-old mice were used for tumor cell inoculation, whereas female 6-to-12-week-old mice were used as BM-cell donors. Conditional gene knockout mice were generated by breeding promoter-specific transgenic *Cre* mice with gene-floxed mice, which were genotyped by polymerase chain reaction according to information provided by JAX or the other providers. All mice were housed in a specific pathogen-free animal facility at the Institute of Molecular Biology (IMB), AS. The animal experiment protocols were approved by the IACUC of AS. EO771 cells (CH3 Biosystems, NY, USA, Product#94A001, RRID:CVCL_GR23) were cultured in RPMI-1640 (Gibco, Grand Island, NY, USA) containing 10% FBS (Hyclone, Marlborough, MA, USA), 20 mM HEPES (Sigma-Aldrich, St. Louis, MO, USA), 100 U/mL penicillin, and 100 μg/mL streptomycin (Gibco).

### 2.2. Construction of BM Chimeras

BM cells isolated from the femurs and tibia of donor mice were depleted of red blood cells (RBCs) using ammonium-chloride-potassium (ACK) buffer (150 mM NH_4_Cl, 10 mM KHCO_3_, 1 mM EDTA), and then incubated with 2.4G2 hybridoma supernatant (produced in-house) to block non-specific binding of antibodies to Fc receptors (FcRs). T and B cells were removed using biotin-conjugated anti-CD90.2 (53-2.1, produced in-house) and anti-B220 (RA3-6B2, BioLegend Cat#103203, RRID:AB_312988, San Diego, CA, USA) antibodies and streptavidin microbeads (Miltenyi Biotec, Bergisch Gladbach, Germany) with an LD column and a QuadroMACS Separator (Miltenyi Biotec). B6 recipient mice received 10 Gy of *γ*-radiation from a ^137^Cs source (J.L. Shepherd & Associates, Sab Fernando, CA, USA) and then an injection of 10^6^ T/B-cell-depleted BM cells via the tail vein. The BM chimeras were used for experiments 8 weeks after being generated.

### 2.3. Tumor Model and CTX Therapy

Mice were inoculated with 0.5 × 10^6^ E0771 cells into the right fourth mammary fat pad on day 0 and then injected intraperitoneally (ip) with 150 mg/kg body weight CTX (Sigma-Aldrich) on day 21 and day 27, unless stated otherwise. For depletion of T cells in vivo, mice received an ip injection of 0.2 mg anti-CD4 (GK1.5, RRID:AB_1107636, BioXcell, Lebanon, NH, USA), anti-CD8*α* (2.43, RRID:1125541, BioXcell), anti-CD4 plus anti-CD8*α*, or isotype control antibody (Rat IgG2b, RRID:1107780, κ; BioXcell) 8 h before each CTX treatment. Tumor volume was measured every 2–3 days starting from 12 days post-inoculation, and it was calculated as length × width^2^ × 0.52. Mice with a tumor volume exceeding 2000 mm^3^ were considered moribund and euthanized. The mean tumor size curves record the average tumor size of all mice in each group before the appearance of any moribund mouse.

### 2.4. Flow Cytometry Analysis of Cells from Dissociated Tumors

Tumors were dissociated into single cells using a Tumor Dissociation Kit (Miltenyi Biotec) according to the manufacturer’s instructions. In brief, the tumor was minced and then incubated in digestion buffer at 37 °C for 40 min with shaking (200 rpm). The resulting cell suspension was passed through a 70-μm strainer (BD Bioscience, Franklin Lakes, NJ, USA), incubated in ACK buffer to remove RBCs, and then used for flow cytometry analysis.

Cells were suspended in 2.4G2 hybridoma supernatant to block FcRs and incubated with a fluorophore-conjugated antibody cocktail (panel details are provided in the Appendix A) in staining buffer (PBS containing 2% FBS and 0.02% NaN_3_) for 15 min at room temperature. For intracellular staining, cells were fixed for 30 min at 4 °C with 4% paraformaldehyde (Sigma-Aldrich) to stain cytosolic molecules or with a Foxp3/Transcription Factor Fixation/Permeabilization Concentrate and Diluent kit (Thermo Fisher Scientific, Waltham, MA, USA) to stain transcription factors. The fixed cells were washed with staining buffer and permeabilized with 0.1% saponin (Sigma-Aldrich) at 4 °C for 30 min. The fluorophore-conjugated antibodies for intracellular staining were prepared in staining buffer containing 0.1% saponin to stain cells at 4 °C for 30 min. Flow cytometry was performed using FACSymphony A3 (RRID:SCR_023644, BD Biosciences), and the data were analyzed in FlowJo software v10.10 (RRID:SCR_008520, BD Biosciences). Antibodies used for flow cytometry analysis of cells dissociated from tumor tissue were listed in Appendix A. 

### 2.5. Statistical Analysis

An unpaired *t*-test was used to compare two experimental groups. A two-way ANOVA was applied to analyze tumor size statistically. The Kaplan–Meier estimator was employed for survival analysis, and statistical significance was determined by a Log-Rank test. All statistical analyses were performed using GraphPad Prism 7 (GraphPad, Boston, RRID:SCR_002798, MA, USA). * *p* < 0.05; ** *p* < 0.01; *** *p* < 0.001; **** *p* < 0.0001. Data are presented as mean ± SEM.

## 3. Results

### 3.1. CTX Therapy Promotes Long-Term Survival of Mice with Advanced EO771 Breast Cancer but Requires CD8^+^ T Cell Immunity

Previous studies have reported that repeated injection of a medium dose of CTX, i.e., 140 or 170 mg/kg body weight, every 6 days inhibited tumor growth in several syngeneic immunocompetent mouse models [4,5,23]. To examine the long-term effect of CTX therapy on advanced cancer, we adopted the same 6-day intermittent treatment schedule comprising two injections of a medium dose of CTX (150 mg/kg) starting at day 21 post-orthotopic inoculation of EO771 breast cancer cells (Figure 1A schema), i.e., when spontaneous lung metastasis had occurred [19]. We found that our CTX regimen induced tumor regression and prolonged overall survival (OS) throughout the follow-up period of up to 120 days post-tumor inoculation (Figure 1A). Notably, the tumors disappeared in a median of 56.7% (42.9–70.6%) of the EO771-bearing mice, indicative of a curative effect of this therapeutic approach on mice displaying metastatic EO771 breast cancer.

Given that CTX modulates T cell activation, we explored if T cells are involved in CTX efficacy in the EO771 model. We found that CTX treatment only induced transient tumor regression but did not promote OS in *Rag2*^−/−^ mice that lack B and T lymphocytes (Figure 1B and Appendix A). Moreover, the depletion of CD8^+^, but not CD4^+^, cells in wild type (WT) mice abolished the therapeutic effect of CTX (Figure 1C). These results indicate that CD8^+^ T cells are essential for CTX’s anti-tumor effect. Given that the activation of CD8^+^ T cells, which recognize cell-associated antigens, requires cross-presentation of the antigens by type 1 conventional dendritic cells (cDC1s), we examined the role of cDC1 using *Batf3*^−/−^ mice that lack cDC1s [24]. We observed that CTX treatment induced transient tumor regression but failed to promote the OS of the *Batf3*^−/−^ mice (Figure 1D and Appendix A). Collectively, these results demonstrate that the two-medium-dose CTX regimen effectively treats advanced EO771 breast cancer, but this outcome depends on CD8^+^ T cells and likely also on cDC1s. The requirement of CD8^+^ T cells for promoting OS in this advanced breast cancer model is consistent with the requirement of CD8^+^ T cells for the suppression of tumor growth in earlier studies using less advanced tumor models [4,5,23].

### 3.2. The Effect of CTX Therapy Requires the IFNar1 and cGAS/STING of Bone Marrow-Derived Cells

CTX induces the production of IFN-I in vivo. The requirement of host response to IFN-I for the augmentation of tumor regression and T cell responses by CTX treatment was demonstrated in previous studies using IFNAR1 blockade with antibody [4] and *Ifnar1*^−/−^ mice [12,13], respectively. To examine the role of the IFN-I response in bone marrow (BM)-derived cells, we generated *Ifnar1*^−/−^ BM chimeras in B6 mice (*Ifnar1*^−/−^ → B6) and found that, unlike the B6 → B6 controls, they succumbed to EO771 tumors despite CTX treatment (Figure 2A and Appendix A). As a CTX metabolite causes the interstrand DNA crosslinks that result in the death of proliferating cells, we hypothesized that the dsDNA released from cells killed by CTX treatment triggers the cytosolic dsDNA-sensing cGAS–STING pathway and consequent IFN-I production by BM-derived cells, presumably in the tumor microenvironment (TME). To determine the role of cGAS and STING in BM-derived cells, we generated *Cgas*^−/−^ and *Sting1^gt/gt^* BM chimeras in B6 mice and found that the pro-survival effect of CTX therapy was significantly reduced in the *Cgas*^−/−^ → B6 chimeras and abolished in almost all *Sting1^gt/gt^* → B6 chimeras (Figure 2B and Appendix A). These results indicate that the IFN-I response and cGAS/STING of BM-derived cells are crucial for the anti-tumor effect of CTX, in which the activation of the cGAS–STING–IFN-I pathway in BM-derived cells presumably plays a pertinent role.

### 3.3. STING and IFNar1 of Distinct Myeloid Cells Are Essential for CTX Efficacy

To delineate the type of BM-derived cells whose STING or IFN-I response contributes to CTX’s anti-tumor effect, we generated promoter-specific *Sting1* or *Ifnar1* deletion mice by crossing a mouse strain with *Cre* driven by the *LysM*, *CD11c,* or *Xcr1* promoter to a mouse strain carrying floxed *Sting1* (*Sting1^f/f^*) or *Ifnar1* (*Ifnar1^f/f^*) and then used their BM cells to construct chimeras in B6 mice. *LysM-Cre* mediates the deletion of floxed genes in macrophages, neutrophils, and some monocytes; *CD11c-Cre* mediates their deletion in cDCs and plasmacytoid DCs; and *Xcr1-Cre* mediates their deletion specifically in cDC1s [25,26]. In terms of STING, we found that the therapeutic effect of CTX had been lost in the *LysM-Cre/Sting1^f/f^* → B6, *CD11c-Cre/Sting1^f/f^* → B6, and *Xcr1-Cre/Sting1^f/f^* → B6 BM chimeras in comparison to the *Sting1^f/f^* → B6 BM chimeras (Figure 3A and Appendix A). These results indicate that cDC1 STING is essential for CTX efficacy, with the STING of certain LysM^+^ myeloid cells, likely representing macrophages, also being required.

With regard to the IFN-I response, we found that the efficacy of CTX was lost in the *LysM-Cre/Ifnar1^f/f^* → B6 and *CD11c-Cre/Ifnar1^f/f^* → B6 BM chimeras but remained intact for the *Xcr1-Cre/Ifnar1^f/f^* → B6 BM chimeras, i.e., to the same extent as determined for the *Ifnar1^f/f^* → B6 BM chimeras (Figure 3B and Appendix A). Thus, the IFN-I response of certain non-cDC1 myeloid cells also appears to be essential for CTX efficacy. IFN-I signaling in cDCs, including that induced by STING agonists, upregulates the expression of IL-15/IL-15R*α* by cDCs, which is thought to augment anti-tumor immunity via the activation of cDCs, CD8^+^ T cells, and natural killer (NK) cells [27,28,29]. Therefore, we examined the role of cDC1-generated IL-15 in CTX efficacy. We uncovered that our two-medium-dose CTX therapy is similarly effective for the *Xcr1-Cre/Il15^f/f^* → B6 BM and *Il15^f/f^* → B6 BM chimeras (Figure 3C and Appendix A), indicating that cDC1-generated IL-15 is dispensable for CTX’s effectiveness. This finding is in line with IFN-I signaling in cDC1s being dispensable for CTX efficacy (Figure 3B).

### 3.4. cGAS and STING of BM-Derived Cells Positively Modulate the CD8^+^ T Cell Response

Given that CD8^+^ T cells (Figure 1B,C) and the cGAS/STING of BM-derived cells (Figure 2) are essential for CTX treatment efficacy, we examined if the cGAS and STING of BM-derived cells affect CD8^+^ T cells under CTX treatment conditions. To do so, we subjected the *Cgas*^−/−^ → B6, *Sting1^gt/gt^* → B6, and B6 → B6 BM chimeric mice bearing EO771 tumors to the CTX regimen, and then their tumors were harvested eight days later to examine their CD8^+^ T cells. We observed that the tumors from either of the mutant BM chimeras harbored similar proportions of CD8^+^ T cells among CD45^+^ cells and PD-1^+^ cells among CD8^+^ T cells as those determined for tumors from their respective WT BM chimeric controls (Appendix A). T cell receptor (TCR) signaling induces PD-1 expression in T cells [30]. We found that the intratumoral PD-1^+^CD8^+^ T cell population consists of both high and low PD-1 expressers (PD-1^hi^ and PD-1^lo^) at a ratio of approximately 9:1, which is not affected by cGAS or STING deficiency in the BM-derived cells (Figure 4A). Chronic stimulation of CD8^+^ T cells by tumor antigens drives T cell exhaustion. Whereas PD-1^lo^ marks T cell activation, PD-1^hi^ indicates progression toward T cell exhaustion [31,32]. Consistently, we found that the intratumoral PD-1^hi^CD8^+^ T cells of our WT and mutant BM chimeras expressed higher levels of the T cell exhaustion markers, Lag-3 and Tim-3, compared to their PD-1^lo^ counterparts (Figure 4B). Further analysis of expression levels of the inhibitory receptors PD-1, Lag-3, and Tim3 by PD-1^hi^CD8^+^ T cells revealed that the levels of PD-1 were similar between the WT and mutant BM chimeras, whereas the levels of Lag-3 and Tim-3 were significantly higher in both of the mutant BM chimeras compared to their WT controls (Figure 4C). This result indicates that cGAS or STING deficiency in BM-derived cells increases the level of inhibitory Lag-3 and Tim-3 expressed by PD-1^hi^CD8^+^ T cells under the CTX treatment.

Antigen-experienced CD8^+^ T cells in the TME comprise various subsets at distinct differentiation stages that rise sequentially—from stem cell-like (T_SCL_) to progenitor exhausted (T_PEX_) and then to terminally exhausted (T_EX_) CD8^+^ T cells [31,32,33]. The levels of intratumoral CD8^+^ T_SCL_ and T_PEX_ cells, but not T_EX_ cells, are positively associated with the efficacy of cancer immunotherapies, including immune checkpoint blockade, cancer vaccination, and adoptive T cell therapy [34,35]. Accordingly, we assessed if the cGAS or STING of BM-derived cells affects these three subsets of intratumoral CD8^+^ T cells under the conditions of CTX treatment. We used PD-1, Lag-3, and Tim-3 to distinguish the T_SCL_ (PD-1^lo^Lag-3^−^Tim-3^−^), T_PEX_ (PD-1^hi^Lag-3^+^Tim-3^−^), and T_EX_ (PD-1^hi^Lag-3^+^Tim-3^+^) subsets, respectively [31] (Appendix A). The expression of CD44, CD62L, Ly108, and granzyme B (GzmB) by these subsets was in line with the known characteristics of these cell populations (Figure 4D) [33,34]. We found that a deficiency of cGAS, but not STING, in the BM-derived cells reduced the population of T_SCL_ cells, whereas a deficiency of either of those molecules increased the T_EX_ cell population among intratumoral PD-1^+^CD8^+^ cells (Figure 4E). Thus, together with the augmented expression of inhibitory receptors, Lag-3 and Tim-3, by intratumoral PD-1^hi^CD8^+^ T cells in both mutant BM chimeras (Figure 4C), the cGAS and STING of BM-derived cells promote the CD8^+^ T cell response under CTX treatment conditions.

## 4. Discussion

A medium dose of CTX induces IFN-I-dependent modulation of dendritic and T cells in vivo and suppresses tumor growth in a CD8^+^ T cell-dependent manner in several murine models, including its curative effect on advanced EO771 breast cancer, as demonstrated in this study. However, the mechanism underlying the immune-mediated anti-tumor effect of CTX is not yet fully understood. In this study, we found that the efficacy of CTX therapy requires the STING of cDC1s and likely macrophages, the IFN-I response of non-cDC1 myeloid cells, and the cGAS of BM-derived cells. Moreover, the cGAS and STING of BM-derived cells positively modulate the intratumoral CD8^+^ T cell response under CTX treatment, as evidenced by the cGAS or STING deficiency in BM-derived cells resulting in an increase in the population of CD8^+^ T_EX_ cells and the expression of Lag-3 and Tim-3 by PD-1^hi^CD8^+^ T cells, whereas cGAS deficiency, but not that of STING, reduces the population of CD8^+^ T_SCL_ cells.

With regard to the role of STING in the anti-tumor effect of CTX, our results indicate that cDC1s and their STING are essential (Figure 1D and Figure 3A). Similar requirements have recently been reported for the induction of anti-tumor T cell responses by particulate polymeric cGAMP [36,37]. These findings imply that the production of IFN-I by cDC1 is mediated through the activation of STING by cGAMP, which is consistent with earlier findings that STING-dependent IFN-I production by CD11c^+^ antigen-presenting cells (APCs) occurs in tumors either spontaneously [38] or in response to radiation [39]. In addition to the STING of cDC1, we found that the STING of LysM^+^ BM-derived cells, presumably macrophages, is also crucial to the effectiveness of CTX treatment (Figure 3A). Notably, tumor cell-derived DNA in intratumoral CD11c^+^ APCs has been reported in an in vivo study [38]. In a study of polymeric cGAMP using cGAMPs incorporated into virus-like particles, the particles were found to preferentially target cDCs and macrophages [36]. Therefore, we speculate that dsDNA released from tumor cells after CTX treatment in vivo also exists in form(s) that preferentially target APCs, which subsequently triggers the APC-intrinsic cGAS–STING pathway that leads to IFN-I production. This hypothesis is supported by the requirement of cGAS from BM-derived cells for the effectiveness of CTX (Figure 2B). APC-specific *Cgas* knock-out mice will be useful to investigate the function of APC cGAS in vivo.

It is generally thought that IFN-I signaling in cDC1s is required for the anti-tumor T cell response. For instance, IFN-I directly promotes the cross-presentation of cell-associated antigens, as well as the activation and survival of cDC1s [40]. Moreover, the spontaneous anti-tumor T cell response in either mixed *Baft3*^−/−^ and *Ifnar1*^−/−^ BM chimeras [41] or *CD11c^Cre^Ifnar1^f/f^* mice [42] was previously shown to be impaired. However, using *XCR1^Cre^Ifnar1^f/f^* mice, we found that the IFN-I response of cDC1s is dispensable for the efficacy of CTX (Figure 3B). Our finding is in line with a recent study reporting that IFN-I signaling in cDC1s is nonessential for the spontaneous tumor control displayed by T cells using *XCR1^Cre^Ifnar1^f/KO^* or *Karma^Cre^Ifnar1^f/f^* mice [43]. The discrepancy between the recent and earlier studies is likely due to XCR-1, but not Batf3 or CD11c, being specifically expressed by cDC1s; Baft3 and CD11c expression by certain non-cDC1 cells likely impacts anti-tumor immunity. This latter possibility is supported by the fact that a transient expression of Baft3 by CD8^+^ T cells at priming is essential for the survival of activated CD8^+^ T cells proceeding to memory cells [44,45] and by the expression of CD11c by cDC2s and certain macrophage populations [26].

The loss of CTX efficacy in *LysM^Cre^Ifnar1^f/f^* and *Cd11c^Cre^Ifnar1^f/f^* mice that we observe (Figure 3B) indicates that the IFN-I response of non-cDC1 myeloid cells, presumably certain type(s) of macrophages and maybe cDC2s, is required for CTX efficacy. Although EO771 cells express IFN-I-induced genes in response to 4-hydroperoxy-CTX in vitro [4], we speculate that the EO771-derived IFN-I in our advanced tumor model is insufficient to support the anti-tumor effect of CTX due to chromosomal instability (CIN) in EO771 cells [46,47]. This is because tumor cells in advanced cancers have adapted to cope with the CIN-induced chronic activation of the cGAS–STING–IFN-I pathway by re-wiring the STING downstream signaling away from IFN-I induction so that IFN-I-mediated immune stimulation is lost [48,49]. Thus, our finding that CTX treatment activates the cGAS–STING–IFN-I axis in BM-derived cells of mice with advanced breast cancer indicates that CTX bypasses the IFN-I defect of tumor cells, implying that CTX can be successfully deployed to treat CIN tumors that account for more than 90% of solid tumors [50].

The cGAS of tumor cells has been shown to play a critical role in anti-tumor immunity, such as in the spontaneous CD8^+^ T cell response against mismatch repair-deficient tumor cells [51] and in the anti-tumor T cell response induced by PARP or checkpoint kinase 1 inhibitors that target the DNA damage response [52,53]. Other studies have found that the sensing of tumor cell-derived cGAMP by the STING of non-tumor cells is crucial to anti-tumor NK cell [54] and CD8^+^ T cell [55] responses, as well as to the immune-mediated curative effect of ionizing radiation [46]. Here, we found that cGAS of BM-derived cells is essential for the efficacy of CTX therapy (Figure 2B). Since EO771 tumor cells are able to generate and export cGAMP spontaneously due to CIN [46,47], this need for host cGAS is interesting and has the following potential implications. One possibility is that most tumor cells are killed shortly after CTX treatment [7] and that the consequent released dsDNA is taken up by host APCs to activate APC-intrinsic cGAS. Another possibility is that the effectiveness of CTX requires a non-canonical cGAMP-independent function of cGAS in BM-derived cells. This latter possibility is in line with our finding that a deficiency of cGAS, but not STING, in BM-derived cells reduces the proportion of CD8^+^ T_SCL_ cell population in tumors (Figure 4E), indicating that a STING-independent cGAS function sustains the level of intratumoral CD8^+^ T_SCL_ cells. STING-independent cGAS functions have been reported recently, including the direct actions of cGAS independently of its cGAMP synthase activity [56,57,58] and cGAMP-dependent functions [59,60]. It remains to be determined if the STING-independent function of cGAS detected in our study involves cGAMP. Given the importance of CD8^+^ T_SCL_ cells in anti-tumor immunity, this novel cGAS function of BM-derived cells may represent a target for immunotherapy.

## 5. Conclusions

The CTX regimen effectively treats mice with advanced EO771 breast cancer. We propose a model for the CD8^+^ T cell-dependent anti-tumor effect of CTX, which acts via cGAS and STING of host myeloid cells (Figure 5). The active metabolite of CTX induces the immunogenic death of proliferating tumor cells via dsDNA crosslinking. The dsDNA released from the dead tumor cells effectively targets APCs and triggers the cGAS–STING–IFN-I pathway. The STING of cDC1s and macrophages and the IFN-I response of certain LysM^+^ or/and CD11c^+^ non-cDC1 myeloid cells are essential for CTX efficacy. Under CTX treatment conditions, the cGAS and STING of BM-derived cells facilitate a CD8^+^ T cell response in tumors by suppressing the levels of CD8^+^ T_EX_ cell population and the Lag-3 and Tim-3 expression by PD-1^hi^CD8^+^ T cells, while the cGAS of BM-derived cells sustains the level of CD8^+^ T_SCL_ cell population independent of STING.

## Figures and Tables

**Figure 1 cancers-17-01130-f001:**
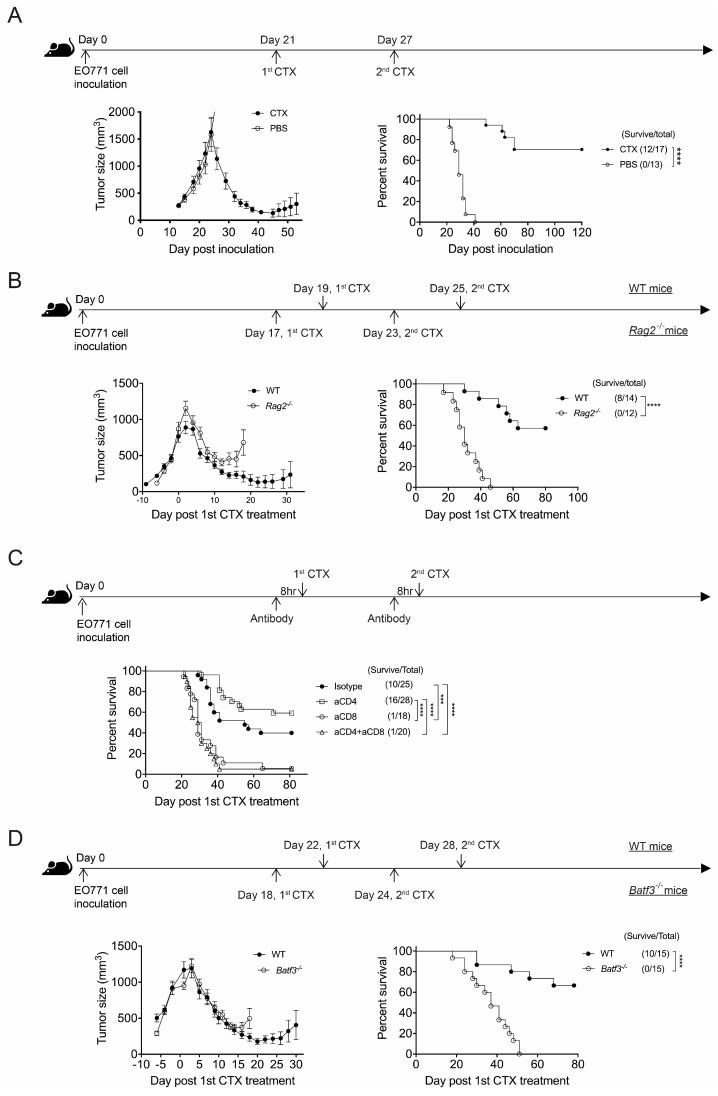
CTX therapy promotes the long-term survival of mice with advanced E0771 breast cancer but requires CD8^+^ T cell immunity. (**A**) CTX effectively cures WT B6 mice having advanced EO771 cancer. B6 mice bearing EO771 tumors were injected with CTX or PBS at day 21 and day 27 post-tumor inoculation and then monitored for tumor volume and survival. Representative data from one of five independent experiments are shown. (**B**) CTX efficacy requires *Rag2*-dependent cells. WT and *Rag2*^−/−^ mice bearing EO771 tumors received the first CTX injection when the average tumor volume had reached 800 mm^3^ and the second CTX injection 6 days later. Tumor volume and survival were monitored. Representative data from one of two independent experiments are shown. (**C**) CTX efficacy requires CD8^+^ cells. EO771-bearing B6 mice were administered the indicated antibodies 8 h prior to each CTX injection. The first CTX treatment was given when the average tumor size had reached 800 mm^3^, and it was followed by the second CTX treatment 6 days later. Mouse survival was monitored every 2–3 days. Data are compiled from two independent experiments. (**D**) Batf3 deficiency abolishes CTX efficacy. WT and *Batf3*^−/−^ mice bearing EO771 tumors received the first CTX injection when the average tumor volume had reached 800 mm^3^ and the second CTX injection 6 days later. Tumor volume and mouse survival were monitored. Representative data from one of two independent experiments are shown. The values in brackets represent n, which is the same for tumor size and % survival. Each tumor volume data point is the mean ± SEM of the indicated mouse group. *** *p* < 0.001; **** *p* < 0.0001.

**Figure 2 cancers-17-01130-f002:**
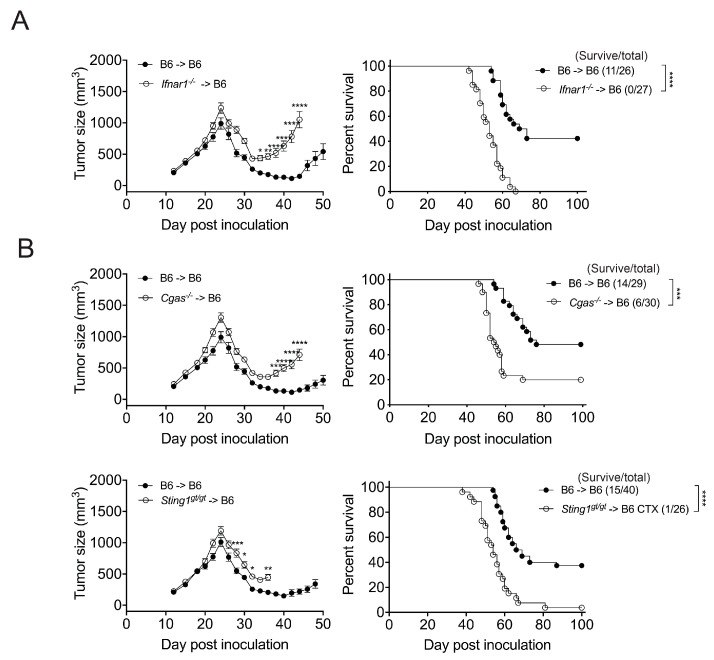
The effect of CTX therapy requires the IFNar1 and cGAS/STING of BM-derived cells. BM chimeric mice bearing EO771 tumors were treated with CTX at day 21 and day 27 post-tumor inoculation and monitored for tumor volume and survival. (**A**) *Ifnar1*^−/−^ → B6 versus B6 → B6 chimeras, (**B**) *Cgas*^−/−^ → B6 versus B6 → B6 chimeras, and *Sting1^gt/gt^* → B6 versus B6 → B6 chimeras. Data are compiled from 2-3 independent experiments. Each experiment has its own B6 → B6 chimera control group. * *p* < 0.05; ** *p* < 0.01; *** *p* < 0.001; **** *p* < 0.0001.

**Figure 3 cancers-17-01130-f003:**
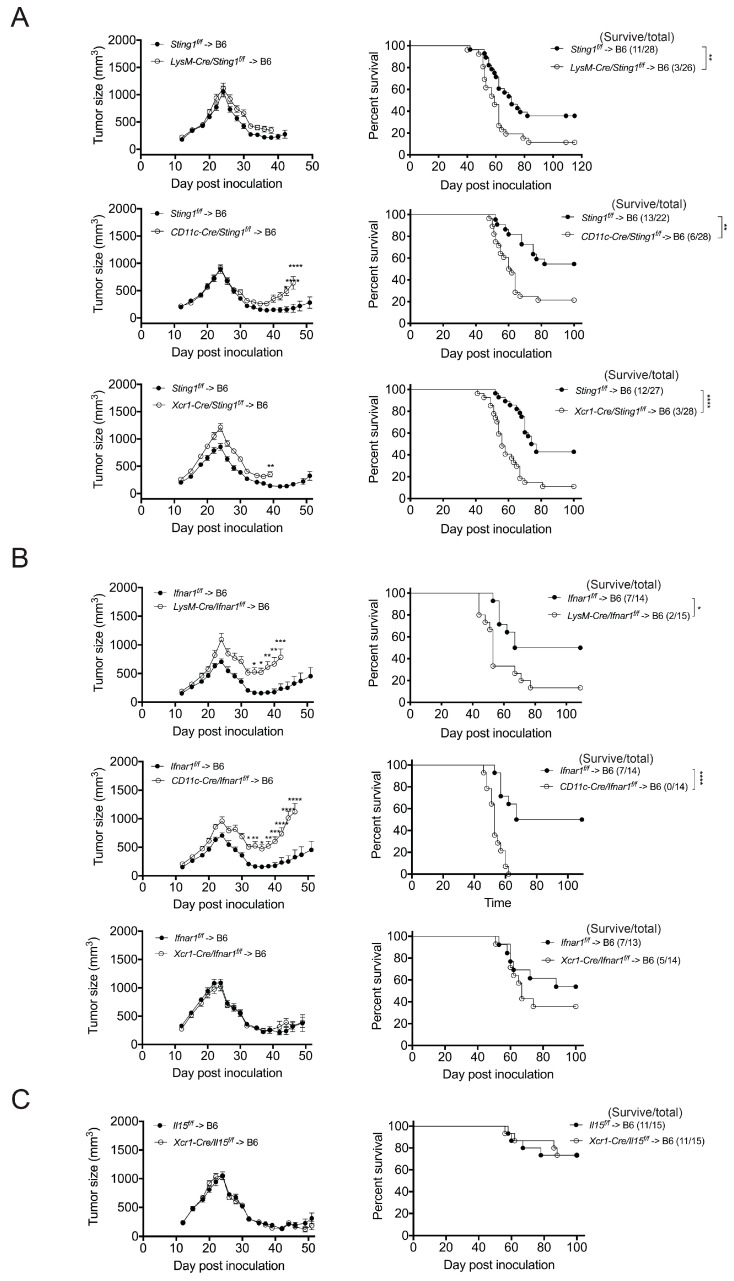
STING and IFNar1 of distinct myeloid cells are essential for CTX treatment efficacy. The experimental design is the same as for Figure 1A. (**A**) STING deficiency in myeloid cells abolishes CTX treatment efficacy. Tumor volume and survival of *LysM^cre/+^Sting1^f/f^* → B6, *CD11c^cre/+^Sting1^f/f^* → B6, and *XCR1^cre/+^Sting1^f/f^* → B6 BM chimeric mice were monitored. Each mutant BM chimeric group has its own *Sting1^f/f^* → B6 BM chimera control. Data have been compiled from 2–3 experiments. (**B**) Loss of IFN-I receptor from non-cDC1 myeloid cells abolishes CTX treatment efficacy. Tumor volume and survival of *LysM^cre/+^IFNar1^f/f^* → B6, *CD11c^cre/+^IFNar1^f/f^* → B6, and *XCR1^cre/+^IFNar1^f/f^* → B6 BM chimeric mice were monitored. Each mutant BM chimeric group has its own *IFNar1^f/f^* → B6 BM chimera control. Representative data from one of two independent experiments are shown. (**C**) IL-15 of cDC1s is dispensable for CTX treatment efficacy. Tumor volume and survival of *Xcr1^cre/+^Il15^f/f^* → B6 and *Il15^f/f^* → B6 BM chimeric mice were monitored. Representative data from one of two independent experiments are shown. Each symbol in the tumor size graphs represents the mean ± SEM of all mice in that group at the indicated time. Statistical significance was determined by two-way ANOVA (for tumor size) or the Log-Rank test (for survival). * *p* < 0.05; ** *p* < 0.01; *** *p* < 0.001; **** *p* < 0.0001.

**Figure 4 cancers-17-01130-f004:**
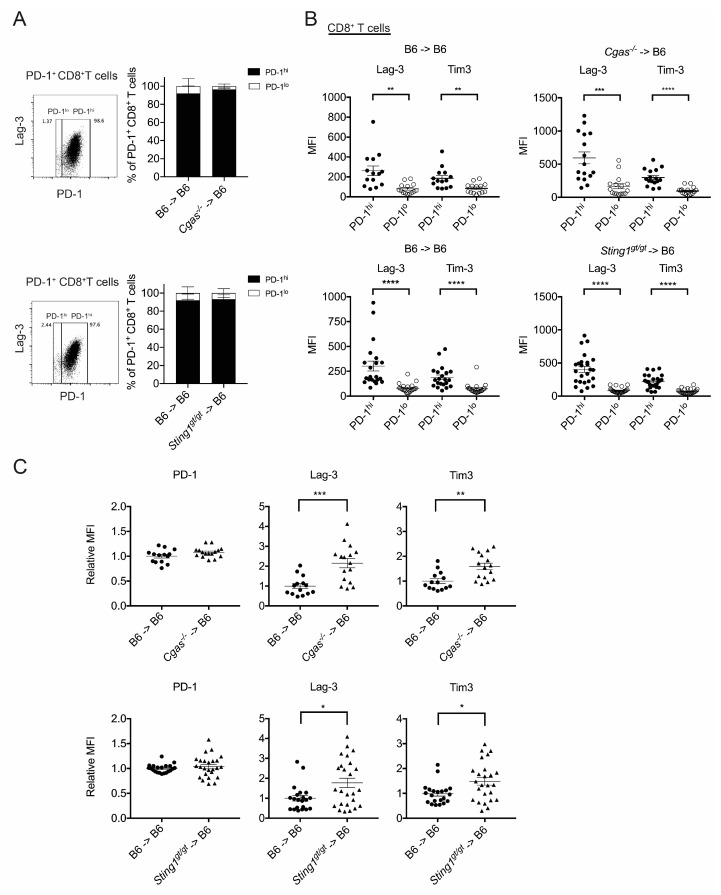
cGAS and STING of BM-derived cells positively modulate the CD8^+^ T cell response. BM chimeric mice bearing EO771 tumors were treated with CTX at day 21 and day 27 post-tumor cell inoculation, and then, the tumors were harvested on day 35 to examine CD8^+^ T cells (Figure 4, Appendix A). (**A**) The composition of PD-1^hi^ and PD-1^lo^ subsets among PD-1^+^CD8^+^ T cells. (**B**) Comparison of the expression of Lag-3 and Tim-3 between PD-1^hi^ and PD-1^lo^ CD8^+^ T cells. MFI, mean fluorescence index. Data have been compiled from two independent experiments. (**C**) Comparison of the expression of PD-1, Lag-3, and Tim-3 by PD-1^hi^CD8^+^ T cells between mutant BM → B6 chimeras and their WT BM → B6 control. Data have been compiled from two independent experiments. Relative MFI was calculated by normalizing the MFI from a sample against the mean MFI of the WT BM → B6 group in each independent experiment. (**D**) Expression of Lag3, Tim3, CD44, CD62L, Ly108, and GzmB by the intratumoral CD8^+^ T_SCL_, T_PEX_, and T_EX_ subsets (Figure 4, Appendix A). Negative controls are either fluorescence-minus-one or the maximum background of the same fluorescence channel from single staining of another molecule. Data from representative *Cgas*^−/−^ → B6 and representative *Sting1^gt/gt^* → B6 BM chimeras are shown. (**E**) Comparison of the proportion of T_SCL_, T_PEX_, and T_EX_ subsets among PD1^+^CD8^+^ T cells between the mutant BM chimeras and their WT controls. Data have been compiled from two independent experiments. Relative % of PD1^+^CD8^+^ T cells was calculated by normalizing the % of a sample against the mean % of the WT BM → B6 group in each independent experiment. Each symbol in the graph represents 1–4 mice from the indicated B6 → B6 chimeric group or one mouse from the indicated mutant BM → B6 chimeric group. Mean ± SEM of all samples in each group is presented. Statistical significance was determined by unpaired two-tailed Student’s *t*-test. Appendix A (related to (**A**)). Appendix A (related to (**D**)). * *p* < 0.05; ** *p* < 0.01; *** *p* < 0.001; **** *p* < 0.0001.

**Figure 5 cancers-17-01130-f005:**
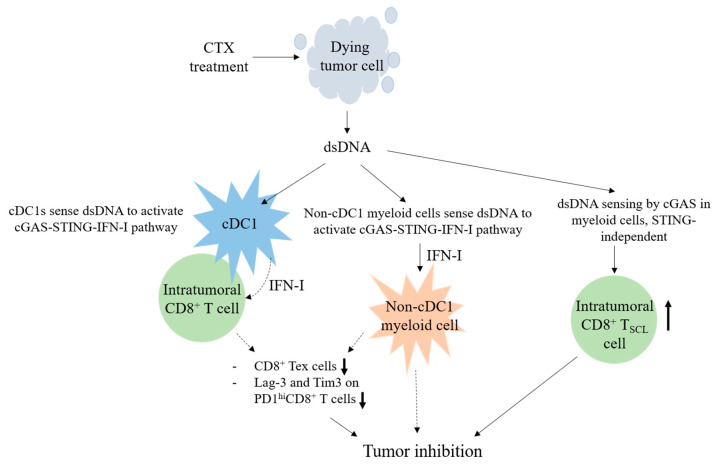
A model for CD8^+^ T cell-dependent anti-tumor effect of CTX therapy via activation of myeloid cGAS and STING. The thick **↑** and **↓** indicate an increase and a decrease, respectively, of the designated feature of intratumoral CD8^+^ T cells. The dashed arrows indicate suggested events.

## Data Availability

The original contributions presented in this study are included in the article/Appendix A. Further inquiries can be directed to the corresponding author.

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
