# Peer review of "cGAS and STING in Host Myeloid Cells Are Essential for Effective Cyclophosphamide Treatment of Advanced Breast Cancer"

_cancers, 2025, doi:10.3390/cancers17071130_

Round 1
Reviewer 1 Report
Comments and Suggestions for Authors
Author comments are in attached pdf

Author Response
Please see the uploaded "Response to Reviewer 1 Comments" file.

Reviewer 2 Report
Comments and Suggestions for Authors
The present study showed the important role of cGAS-Sting in CD8 T cells and cDCs cooperatively suppressing breast cancer by CTX treatment. The experimental data are well done and sufficient to support the authors' conclusions. I suggest that authors could draw a graphic summary to let readers easy to understand how CD8 T cells and cDCs suppressed tumor growth by CTX treatment.
Author Response
Please see the uploaded "Response to Reviewer 2 Comments" file.

Reviewer 3 Report
Comments and Suggestions for Authors
This study examines how the cGAS-STING-IFN-I pathway influences cyclophosphamide (CTX) treatment for advanced breast cancer. The authors provide solid evidence that cGAS and STING in host myeloid cells play a key role in CTX-induced immune responses, particularly in activating type 1 conventional dendritic cells (cDC1s) and IFN-I signaling in myeloid cells. These findings improve our understanding of CTX’s immune mechanisms and could help refine cancer treatments. I think the manuscript can be accepted in its present form.
Author Response
Please see the uploaded "Response to Reviewer 3 Comments" file.

Round 2
Reviewer 1 Report
Comments and Suggestions for Authors
The authors have provided a sufficient response to the pior review. The manuscript is now suitable for acceptance and publication.
Comments on the Quality of English Language
Ok